# Galanin System in the Human Bile Duct and Perihilar Cholangiocarcinoma

**DOI:** 10.3390/cells12131678

**Published:** 2023-06-21

**Authors:** Sara Huber, Theresia Fitzner, René G. Feichtinger, Sarah Hochmann, Theo Kraus, Karl Sotlar, Barbara Kofler, Martin Varga

**Affiliations:** 1Research Program for Receptor Biochemistry and Tumor Metabolism, Department of Pediatrics, University Hospital of the Paracelsus Medical University, 5020 Salzburg, Austria; sar.huber@salk.at (S.H.); theresia.fitzner@stud.pmu.ac.at (T.F.); 2Department of Pediatrics, University Hospital of the Paracelsus Medical University, 5020 Salzburg, Austria; r.feichtinger@salk.at; 3Cell Therapy Institute, Spinal Cord Injury and Tissue Regeneration Center Salzburg (SCI-TReCS), Paracelsus Medical University, 5020 Salzburg, Austria; sarah.hochmann@pmu.ac.at; 4Department of Pathology, University Hospital of the Paracelsus Medical University, 5020 Salzburg, Austria; t.kraus@salk.at (T.K.); k.sotlar@salk.at (K.S.); 5Department of Surgery, University Hospital of the Paracelsus Medical University, 5020 Salzburg, Austria; m.varga@salk.at

**Keywords:** galanin, galanin receptor, peptide, bile duct, cholangiocytes, cholestasis, cholangiocarcinoma

## Abstract

Background: Perihilar cholangiocarcinoma (pCCA) is characterised by poor outcomes. Early diagnosis is essential for patient survival. The peptide galanin (GAL) and its receptors GAL_1–3_ are expressed in various tumours. Detailed characterisation of the GAL system in pCCA is lacking. Our study sought to characterise GAL and GAL_1–3_ receptor (GAL_1–3_–R) expression in the healthy human bile duct, in cholestasis and pCCA. Methods: Immunohistochemical staining was performed in healthy controls (*n* = 5) and in the peritumoural tissues (with and without cholestasis) (*n* = 20) and tumour tissues of pCCA patients (*n* = 33) using validated antibodies. The score values of GAL and GAL_1–3_–R expression were calculated and statistically evaluated. Results: GAL and GAL_1_–R were expressed in various bile duct cell types. GAL_2_–R was only slightly but still expressed in almost all the examined tissues, and GAL_3_–R specifically in cholangiocytes and capillaries. In a small pCCA patient cohort (*n* = 18), high GAL expression correlated with good survival, whereas high GAL_3_–R correlated with poor survival. Conclusions: Our in-depth characterisation of the GAL system in the healthy human biliary duct and pCCA in a small patient cohort revealed that GAL and GAL_3_–R expression in tumour cells of pCCA patients could potentially represent suitable biomarkers for survival.

## 1. Introduction

Cholangiocarcinoma (CCA) is a rare tumour accounting for 2% of all malignancies with increasing mortality rates [1,2,3]. CCA arises from the ductal epithelium of the biliary tree and is classified in accordance with its localisation as either intra- or extrahepatic, which is then further subdivided into perihilar and distal cholangiocarcinoma [2,3,4]. Perihilar cholangiocarcinoma (pCCA) is anatomically defined as a biliary tree tumour located between the origin of the cystic duct and the liver [2,3]. It accounts for approximately 60% of CCA [1,5,6,7]. pCCA is often diagnosed at advanced stages, and consequently, is characterised by poor clinical outcomes with 1- and 3-year survival rates of 26.2% and 3.4%, respectively. Therefore, early diagnosis would be essential for treatment success [5,6].

It is known that in the pathogenesis of CCA, various signalling pathways are associated with the activation of tumour onset and progression [2,3,8,9]. Moreover, recent studies have revealed that the neuroendocrine system, in particular, plays an important role in tumour progression [10]. For example, in CCA, serotonin promotes sustained cell proliferation [11], melatonin promotes the regression of apoptosis [12] and oestrogen promotes angiogenesis [13]. These and other neuroendocrine hormones are similarly expressed in cholestasis [14].

In the present study, we focused on galanin (GAL), a regulatory peptide that was discovered by Viktor Mutt and Kazuhiko Tatemoto in 1983 and originally isolated from the porcine intestine [15]. Alongside its three receptors (GAL_1–3_-R), galanin also belongs to the neuroendocrine system [16]. The amino acid sequence of GAL is highly conserved among numerous species, from fish to human [15,16,17]. Human GAL (30 amino acids) originates from a larger precursor peptide (123 amino acids) encoded by the GAL gene on chromosome 11q13.2 [16,18,19]. The transcription of GAL is regulated by signalling with downstream activation of protein kinase A and Ca^2+^-dependent protein kinase C. [16,20,21]. The galanin receptors belong to the G-protein-coupled receptor (GPCR) family containing the classical seven transmembrane domains, which are intracellularly linked to G-protein subtypes [22,23]. Due to coupling to different G-proteins, downstream intracellular signalling differs between the three receptors [16], although each receptor has overlapping structural features.

GAL acts as a modulator of neurotransmission in the brain and the peripheral nervous system [16]. Furthermore, GAL and its receptors are expressed in various tumours and can modify tumour activity depending on the tumour type, as well as the type of receptor expressed [16,24,25,26,27,28,29,30]. For example, colorectal cancer patients exhibited increased levels of GAL in serum and colon tissues [31,32,33,34], whereas the silencing of GAL and GAL_1_-R induced apoptosis and enhanced the effect of chemotherapy [30]. Reduced GAL_2_-R mRNA levels were observed in prostate, colorectal and breast cancer [35], while low expression of GAL_3_-R was reported to correlate with enhanced metastasis and lymph node invasion in colorectal cancer [31,32]. However, the role of the GAL system in cancer is still not entirely elucidated, as there is evidence for this system serving both tumour promotor and suppressor functions [36].

Cholangiocytes are epithelial cells lining the inner side of the bile duct and are morphologically characterised by a cuboidal shape in small ductules that develop into a columnar shape with progressive enlargement of the ducts, which, ordinarily, are mitotically dormant [37,38,39,40]. Sympathetic and parasympathetic nervous plexuses, as well as rich vascular networks in the bile duct wall, play a role in multiple physiological processes [37,38,39,40]. Peribiliary glands, connective tissue, smooth muscles and adipocytes are further components of the typical bile duct structure. Cholangiocytic proliferation can be induced via bile duct obstruction with cholestasis and is regulated by a complex interaction of multiple factors [41]. One of these factors is GAL, which is increased in serum and liver after bile duct obstruction in rats [42].

Considering the diverse role of the GAL system in cancer [36], and its mitogenic effect on cholangiocytes [42,43], it is conceivable that the GAL system is involved in the carcinogenesis of CCA. Thus, GAL system expression could represent possible markers for the early diagnosis of CCA and survival, and could be potential targets for novel therapy options [36]. Until now, however, the expression of the GAL system in both the healthy human biliary tree as well as in CCA has not been described. This knowledge, in turn, will inform future studies aiming to uncover reliable markers for diagnosis and novel therapeutic approaches.

The aim of the present study was to provide the first detailed overview of cell type-specific expression patterns of GAL and its receptors in the human bile duct via immunohistochemistry with extensively validated antibodies [44]. Further, we elucidated whether the cellular expression of GAL and its receptors is altered in cholestasis. Finally, we defined the cellular pattern of GAL and its receptors in tumour cells of pCCA.

## 2. Materials and Methods

### 2.1. Ethics

This study followed the regulations of the Austrian Gene Technology Act and the Helsinki Declaration of 1975 (revised 2013), and was conducted under approval from the Salzburg State Ethics Research Committee as a non-clinical drug trial or epidemiological investigation. It is registered under the number 1102/2020. All patients gave written informed consent, with anonymity maintained via the codification of patient material.

### 2.2. Sample Characteristics

To characterise the GAL system in the human bile duct, we conducted immunohistochemical stainings on human bile duct sections with extensively validated antibodies [44].

As there is a paucity of information to date regarding the GAL system in the biliary tree, we included an analysis of five bile duct samples derived from healthy individuals (Control; Appendix A), as well as thirty-three histologically evaluated tumour samples derived from patients with pCCA (Appendix A). pCCA samples were surgically obtained between 2007 and 2019 at the Department of Surgery, University Hospital, Paracelsus Medical University of Salzburg, and were provided for the experiment along with corresponding peritumoural bile duct tissues (PITs; Appendix A). PITs refer to tissue samples taken furthest from the tumour, and were used to determine whether all the tumour material had been removed by the surgery. They can also refer to the resection margin of the bile duct, and were examined by a pathologist. PITs were further subdivided into PIT with cholestasis (*n*= 10; PIT+C) and PIT without cholestasis (*n* = 10; PIT-C) to examine the impact of cholestasis on the expression of the GAL system (Appendix A and Table 1).

### 2.3. Clinical Parameters

The following clinical parameters were evaluated: age; sex; survival time; clinical signs of cholestasis and cholangitis; TNM (tumour, node and metastasis) classification; infiltration of veins, nerves and lymphatic vessels; grading; residual tumour after surgery; and Bismuth and UICC classification (Appendix A).

### 2.4. Immunohistochemical Staining (IHC)

Tissue sections were washed with xylol, followed by isopropanol and dH_2_O. After incubation with dH_2_O, slides were placed in antigen retrieval buffer (1 mM EDTA, 10 mM Tris; pH = 9) for 40 min at 95 °C. After cooling for 20 min at RT, slides were transferred to dH_2_O and washed for 3 × 3 min in PBS-T (1x PBS + 0.05% Tween^®^ 20 detergent) to remove residual buffer. For blocking, each slide was covered with Dako REAL™ peroxidase blocking solution (Agilent Technologies, Glostrup, Denmark) and incubated for 10 min. Afterwards, slides were again washed in PBS-T, followed by incubation with the 1st antibody (Ab) diluted in Dako antibody diluent with background reducing components (Table 2). Primary Abs were previously tested for specificity and extensively validated by the laboratory [44]. The 2nd Ab only (negative) controls were alternatively covered with antibody diluent only. Following 40 min of incubation at 37 °C, slides were washed in PBS-T. Subsequently, each slide was covered with the 2nd Ab, using Dako EnVision + System-HRP labelled polymer anti-rabbit solution (Agilent Technologies, Glostrup, Denmark) for 30 min and washed again for 3 × 3 min in PBS-T. For staining, slides were incubated with the peroxidase substrate 3,3′-Diaminobenzidin (DAB) for 10 min to provide comparable results. Staining was terminated by immerging the slides in tap water for 2 × 2.5 min. Subsequently, slides were immersed in Mayer’s haemalum solution for 5 min, briefly rinsed in tap water and dipped twice in 3% HCl-ethanol. Finally, tissue sections were rinsed 10–15 times in increasing concentrations of isopropanol (70% → 96% → 100%) and subsequently in 100% xylol. Sections were mounted with cover slides and dried at RT until hardening of the mounting medium (Histokitt No. 1025/500, Karl Hecht GmbH & Co. KG, Sondheim, Germany) occurred.

Negative controls (Appendix A) and appropriate positive control sections (Appendix A), selected for each antigen of interest, were included in immunohistochemical staining (GAL or GAL_1–3_-R-overexpressing cell lines) [44].

### 2.5. Bright Field Microscopy and Slide Scanner

Bright field microscope images were acquired using the VS-120-L Olympus© slide scanner 100-W system. The settings were adjusted to 20x magnification of one z-scan layer. The scan focus points were set manually with settings adjusted to “focusing samples only” and maximum sample detection sensitivity. Scans were used for image analysis.

### 2.6. Image Analysis and Scoring

Analysis was performed using Olympus OlyVia (V3.8) software (Olympus Soft Imaging Solutions, Tokio, Japan). Each section was analysed for the expression of GAL and GAL_1–3_-R via evaluation of staining intensities in cholangiocytes, connective tissue layers, endothelium, nerve fibres, smooth muscle and adipocytes. This was conducted using a previously described scoring system [45] by scaling the staining intensity from 0–3 (0 = no staining, 1 = mild intensity, 2 = moderate intensity, 3 = strong intensity) and counting the percentage of positive cells with minimum values of ≤5% and maximum values of ≥95%. Final score values were derived by multiplying the intensity and percentage: score value [S] = intensity [I] x percentage [E]. Two analysts performed the evaluation and competed scoring independently. Final score values were calculated, and mean values of both evaluations were used for further statistical analysis (Appendix A).

### 2.7. Statistical Evaluation

Using GraphPad Prism software version 9.0.0 (GraphPad Software©, San Diego, CA, USA), the statistical description of peptide and receptor expression was performed by calculating the mean ± SEM score values from IHC for each tissue structure and antigen. Differences between the two unpaired groups were calculated to determine significance using the Mann–Whitney-test. Multiple-comparison tests were used to compare pCCA-derived with healthy- and PIT-derived cholangiocytes, applying the Kruskal–Wallis-test and post hoc Dunn’s multiple comparison test. *p* < 0.05 was considered statistically significant. Pearson correlation was applied to analyse potential associations between parameters. For analysis of survival, Kaplan–Meier curves were used.

## 3. Results

The control tissue sections showed intact structures without destruction of the bile duct epithelial layer or other histological alterations. By contrast, the peritumoural (PIT) samples showed an infiltration of immune cells, an accumulation of adipocytes and damage to the epithelial layer with galled cholangiocytes, especially in samples with cholestasis. As expected, the tumour tissue sections showed destroyed structures of the bile duct wall that formed irregular gland-like structures with mucin-producing cancer cells and disordered tissue infiltration. An investigation of intact histological structures was impossible in these sections.

### 3.1. Galanin in Bile Duct Tissue

In general, the GAL peptide was expressed in various cell types, including cholangiocytes, muscle cells, nerve fibres, adipocytes and endothelial cells. Therefore, GAL was localised to the plasma membrane and cytoplasm of the various cell types (GAL expression and corresponding score values are depicted in Figure 1 and Figure 2, and Appendix A). A detailed overview of the localisation of GAL expression can be found in Table 3. In many sections, GAL revealed diffuse immunostaining with weak positive background signals. This might be due to the secretion of GAL from GAL-expressing cells [46,47]. The GAL expression results for Control, PIT+C and PIT-C, including all comparisons and associated *p* values, are summarised in Table 3.

#### 3.1.1. Cholangiocytes

High GAL expression was found in cholangiocytes, specifically lining the bile duct lumen (Figure 1A–C and Figure 2A). GAL expression did not differ between healthy tissue and PIT with or without cholestasis (score values of Control = 195 ± 11; PIT-C = 209 ± 18; PIT+C = 205 ± 13).

#### 3.1.2. Arteries, Veins and Capillaries

In healthy tissues, the arterial endothelial cells were positive for GAL in all layers of the connective tissue. By comparison, GAL-like immunoreactivity in PIT with or without cholestasis (PIT+C and PIT-C) was reduced by 54–85% in the arterial endothelium compared to control tissue (artery in mucosa: PIT-C: *p* = 0.0420; PIT+C: *p* = 0.0025; artery in adventitia: PIT+C: *p* = 0.0051) (Figure 1D–F). In cholestasis patients in particular, GAL expression was reduced by more than 78% in mucosal arteries (PIT+C, *p* = 0.0025) and by 85% in arteries within the tunica adventitia (PIT+C, *p* = 0.0051) (Figure 2B).

The endothelial cells of venous and capillary blood vessels showed similar results as arteries compared to the healthy control tissues with highest GAL staining intensities (Figure 1G–I). Like in arteries, GAL expression in PIT both with and without cholestasis (PIT+C and PIT-C) was lower within mucosal veins compared to the healthy controls (PIT+C: 40%, *p* = 0.0303; PIT-C: 36%; *p* = 0.0383). In the tunica adventitia, GAL expression in the veins was, on average, 75% lower in cholestasis patients compared to the healthy controls (*p* = 0.0051, Figure 2B).

The IHC staining of the capillary endothelium is depicted in Figure 1J–L. GAL expression was reduced by 56% in capillaries of the tunica adventitia in cholestasis (*p* = 0.0152, Figure 1J–L and Figure 2B). A trend towards lowered mean expression was also found in the mucosal capillaries of cholestasis patients.

#### 3.1.3. Muscle Cells, Nerve Fibres, Adipocytes and Connective Tissue

Several cell types were positive for GAL in healthy tissues, without significant alterations in PIT+C and PIT-C (Figure 2A). For example, biliary muscles exhibited GAL expression in healthy tissue, which, on average, tended to be lower in PIT tissue (PIT-C: 61% and PIT+C: 46%). In biliary nerve fibres, a trend towards increased GAL expression was observed in large (PIT-C: 56%, PIT+C: 60%) and small (PIT-C: 21%, PIT+C: 29%) nerve fascicles in PIT with and without cholestasis. Adipocytes did not show any altered GAL expression in PITs compared to the healthy controls.

### 3.2. Galanin 1 Receptor—GAL_1_-R

IHC analysis revealed that GAL_1_-R was expressed in cholangiocytes, nerve fibres, adipocytes and blood vessels (Figure 3 and Appendix A). According to its membrane-bound characteristics, receptor staining exhibited no notable background signal and was restricted to the cell membrane and cytoplasm of the various cell types. The localisation of GAL_1_-R expression, as well as the results for Control, PIT+C and PIT-C, including all the comparisons made and the *p* values, are summarised in Table 3.

#### 3.2.1. Cholangiocytes

Cholangiocytes showed moderate IHC staining for GAL_1_-R in healthy tissues and PIT (Figure 3A–C). GAL_1_-R was localised in cholangiocytes towards the apical membrane. Although mean GAL_1_-R expression levels were more than twice as high in cholestasis, the differences did not reach statistical significance in comparison to the control tissue, especially when considering the high variability of GAL_1_-R IHC staining (Figure 4A).

#### 3.2.2. Arteries, Veins and Capillaries

Arteries, veins and capillaries revealed low-to-moderate GAL_1_-R IHC staining (Figure 3D–F). GAL_1_-R was virtually undetectable in the arteries of PIT-C (Figure 3E and Figure 4B). In cholestasis, GAL_1_-R expression showed a trend toward reduction in the mucosal arteries compared to the healthy controls. Neither observation reached statistical significance due to the high GAL_1_-R staining variability in healthy tissues (Figure 4B).

In veins, GAL_1_-R showed similar alterations to arteries, with nearly undetectable levels in PIT-C (Figure 3G–I). In contrast to arteries, this effect was only observed in veins embedded in the tunica adventitia. GAL_1_-R expression in capillaries of the tunica adventitia also tended to be lower in PIT-C and PIT+C compared to the control tissues (Figure 4B).

#### 3.2.3. Muscle Cells, Nerve Fibres, Adipocytes and Connective Tissue

Myocytes, fibroblasts and connective tissues did not show any GAL_1_-R immunoreactivity (Figure 4A). Nerve fibres embedded in small nerve fascicles expressed low levels of GAL_1_-R, with similar staining intensities between all groups (Figure 4A). In large nerve fascicles, PIT-C exhibited a trend towards higher expression compared to the control. Of note, the immunoreactivity of GAL_1_-R accumulated in the adipocytes of PIT (Figure 3M–O). Compared to controls, adipocytes in both PIT groups (PIT-C and PIT+C) showed 5-times higher GAL_1_-R expression, with significance for the cholestasis group (*p* = 0.0341) (Figure 4A).

### 3.3. Galanin 2 Receptor—GAL_2_-R

GAL_2_-R showed overall weak staining intensities in the biliary tree. Its expression was found in cholangiocytes, adipocytes and only at low levels in endothelial cells, but was restricted to the cell membrane in the various cell types (Figure 5). The score values of GAL_2_-R expression are provided in Appendix A. The localisation of GAL_2_-R expression, as well as the results for Control, PIT+C and PIT-C, including all the comparisons made and the *p* values, are summarised in Table 3.

#### 3.3.1. Cholangiocytes

In the healthy control tissue, GAL_2_-R exhibited low immunoreactivity in cholangiocytes, lining the bile duct lumen (Figure 5A). Similar to GAL_1_-R, immunostaining was restricted to the cell membrane, apical towards the duct lumen. In PIT, the cholangiocellular expression of GAL_2_-R tended to be lower in cholestasis, in comparison to the control tissue (Figure 5B,C and Figure 6A).

#### 3.3.2. Arteries, Veins and Capillaries

Arteries, veins and capillaries were otherwise negative for GAL_2_-R, with only slightly positive cells in the veins and capillaries of healthy tissues. The vessels of the peritumoural tissue GAL_2_-R were nearly undetectable in IHC (Figure 5D–L and Figure 6B).

#### 3.3.3. Muscle Cells, Nerve Fibres, Adipocytes and Connective Tissue

GAL_2_-R was detected in muscle cells and nerve fibres in cholestasis, as well as in adipocytes, but predominantly in PIT without cholestasis (no significance; Figure 6A). In connective tissues and fibroblasts, immunoreactivity for GAL_2_-R was detected in the control tissue, as well as in PIT with and without cholestasis (Figure 6A).

### 3.4. Galanin 3 Receptor—GAL_3_-R

GAL_3_-R showed very specific expression in the biliary tissues (Figure 7 and Appendix A) restricted to the cell membrane. The score values of GAL_2_-R expression are provided in Appendix A. The localisation of GAL_3_-R expression, as well as the results for Control, PIT+C and PIT-C, including all the comparisons made and the *p* values, are summarised in Table 3.

#### 3.4.1. Cholangiocytes

The cholangiocellular expression of GAL_3_-R was present in the control and PIT, with low immunoreactivity in all groups (Figure 7A–C). In comparison to the control tissues, GAL_3_-R expression showed a trend towards lower levels in PIT-C and similar levels to the controls in PIT+C (Figure 8A).

#### 3.4.2. Arteries, Veins and Capillaries

Endothelial cells lining arteries and veins exhibited no immunoreactivity (Figure 7D–I and Figure 8B). In capillaries, GAL_3_-R-positive endothelial cells were observed in the healthy and PIT samples, and were localised in the mucosal and muscular bile duct layers (Figure 7J–L). In comparison to the control, GAL_3_-R expression levels were reduced, on average, by 81% (*p* = 0.0030) in PIT-C and by 68% (*p* = 0.0326) in PIT+C compared to the healthy controls (Figure 8B). Capillaries localised in the tunica adventitia were negative for GAL_3_-R.

#### 3.4.3. Muscle Cells, Nerve Fibres, Adipocytes and Connective Tissue

Myocytes, neurons, adipocytes and fibroblasts did not show any immunoreactivity for GAL_3_-R expression (Figure 8A).

### 3.5. The Galanin System in pCCA

To monitor the GAL system in pCCA, sections of pCCA were analysed via IHC. For this analysis, sections of PIT, PIT-C and PIT+C were merged. Since the bile duct wall tissues were destroyed due to tumour infiltration, the evaluation of expression was performed on tumour cells only and compared to the healthy control and corresponding PITs (Figure 9 and Figure 10 and Appendix A). Localisation of the expression of the GAL system, as well as the results for Control, PIT and pCCA, including all the comparisons made and the *p* values, are summarised in Table 4.

#### 3.5.1. Galanin

GAL was highly expressed in tumour cells of pCCA (Figure 9C). Therefore, peptide expression was comparable to healthy control samples with >95% GAL- positive cells (Figure 9A and Figure 10A). Between PIT and CCA tissue, which were derived from the same patient, no distinct alterations in immunoreactivity were observed (Figure 9B and Figure 10). GAL expression was detected in the plasma membrane and in the cytoplasm in all study groups (Table 4).

#### 3.5.2. GAL_1_-R

Expression of GAL_1_-R was observed in healthy control, peritumoural and pCCA tissues (Figure 9D–F). The expression of GAL_1_-R was similar between control and cancer tissues (score values: healthy control = 40 ± 11, pCCA = 37 ± 6) but doubled in PIT (Figure 10B). In comparison to pCCA, GAL_1_-R expression was significantly higher (*p* = 0.0459) in PIT (score values: PIT = 75 ± 13). GAL_1_-R expression was detected in the plasma membrane and cytoplasm in all study groups (Table 4).

#### 3.5.3. GAL_2_-R

Weak GAL_2_-R expression was detected in healthy control tissue, with slightly lower expression in peritumoural and cancer tissues (Figure 9G–I and Figure 10C). GAL_2_-R expression in PIT and pCCA samples also revealed no significant differences. GAL_2_-R expression was detected in the plasma membrane in all study groups (Table 4).

#### 3.5.4. GAL_3_-R

GAL_3_-R was found only in cholangiocytes of the healthy control tissue, with weak immunoreactivity localised apical to the cell membrane (Figure 9J). The immunoreactivity of GAL3-R was barely notable in cholangiocytes of PIT and tumour cells of pCCA tissues (Figure 9K,L). Compared to the healthy controls, GAL_3_-R expression in the pCCA tissue was not only significantly reduced but almost absent in tumour cells of pCCA (*p* = 0.0120, Figure 10D). Where present, GAL_3_-R expression was detected in the plasma membrane in all study groups (Table 4).

### 3.6. Correlation of Galanin and Receptor Expression with Clinical Features

Cases were divided into those with high and low expression of GAL or GAL_1–3_-R tumour cells, respectively. Cases of high expression were defined as having staining intensities above the mean expression in tumour cells of pCCA. Kaplan–Meier survival analysis revealed in a small patient cohort (*n* = 18), where survival data were available, that high-intensity GAL staining was associated with patient survival (*p* = 0.0286, Figure 11A). This is supported by the finding that GAL expression tended to inversely correlate with tumour grading (Figure 11B). In contrast, GAL_3_-R exhibited an inverse correlation with survival rates (*p* = 0.0182, Figure 11C). The comparison of mean GAL_3_-R expression in this small cohort among tumour grades did not show any significant difference (Figure 11D). GAL_1_-R and GAL_2_-R expression correlated with neither patient survival nor tumour grading in the respective cohorts (Appendix A).

## 4. Discussion

GAL was previously described in rodent cholangiocytes to be associated with biliary cell proliferation [42,43]. In human cholangiocytes, we observed high levels of GAL expression. Unlike McMillin et al. [42], we did not observe an increase in cholangiocellular GAL expression in cholestasis. In the mentioned study, the proliferative effect of GAL in rodents was mediated by the activation of GAL_1_-R, which represented the near-exclusively expressed GAL receptor in cholangiocytes [42,43]. We observed that all GAL receptors were expressed by cholangiocytes in healthy human tissue, in particular, GAL_1_-R. However, the expression patterns of receptors changed in cholestatic tissue, with slight downregulation of GAL_2_-R and a concurrent increase in GAL_1_-R, making GAL_1_-R the major receptor in cholangiocytes in cholestasis. Although this upregulation of GAL_1_-R was not statistically significant, such a switch in receptor expression could lead to enhanced cholangiocellular proliferation through activation of the mitogenic ERK1/2 signalling pathway [42]. With the simultaneous production of GAL and GAL receptor expression, our results further support the assertion of GAL signalling operating via an autocrine mechanism [42,43]. Furthermore, we found GAL-producing nerve fibres in biliary nerve fascicles, as well as in smooth muscle layers embedded in the connective tissues of the bile duct wall. Similarly, GAL was previously observed in the nerve terminals of the smooth muscles and submucosal ganglia of the intestine, and was shown to regulate bowel motility [48,49]. This leads to the suggestion that GAL might be involved in the regulation of bile duct motility.

Regarding the distribution of GAL_3_-R in the biliary tree, we found low, but specific, expression in the cholangiocytes and endothelial cells of capillaries. It is assumed that GAL_3_-R abates immune responses in colitis [50]. In PIT (with or without cholestasis), we detected significantly lower expression of GAL_3_-R in endothelial cells in capillaries in comparison to healthy tissue. Remarkably, this was also observed in tumour cells of pCCA in comparison to the controls, suggesting that the suppression of GAL_3_-R may play a role in CCA development. The immunoexpression of GAL and GAL_1–3_-R in healthy controls and PIT was restricted to the plasma membrane and the cytoplasm of cholangiocytes; epithelial cells of the mucosa, muscularis, adventitia, smooth muscle cells and nerve bundles; and endothelial cells of arteries, veins and capillaries, as well as of adipocytes. Similar findings were previously reported for tumour samples of colorectal carcinoma patients [31]. In CCA, however, we only evaluated tumour cells but obtained similar results for this tissue type.

GAL and GAL-Rs play diverse roles in tumour development and proliferation [51]. For example, in squamous cell carcinoma, high GAL levels are associated with poor survival outcome and cancerogenic immune escape, as well as accelerated cancer progression [46,52]. In gastric cancer, however, studies associate GAL with anti-tumour effects, as the downregulation of GAL correlates with metastasis and cancer progression [53]. In the present study, we found a trend towards an inverse correlation of GAL expression in the cholangiocytes of pCCA with tumour grading. Moreover, it is reported that the hypermethylation of GAL promotor regions epigenetically silences GAL in gastric cancer cells, negatively correlating with tumour size and tumour-suppressive properties [54]. In healthy tissues, distal to the gastric tumour, neuronal secreted GAL was reported to enhance cell survival and protect benign cells from caspase-mediated cell death [55].

To the best of our knowledge, our study is the first to determine the GAL system in pCCA, which, histologically, is an adenocarcinoma derived from dysplastic cholangiocytes [37,38,39,40]. The tumour breaches the bile duct wall and grows into the surrounding tissue, forming irregular gland-like structures [56,57]. In our study, tumour cells of pCCA demonstrated high GAL immunoreactivity, with similar mean expression in comparison to healthy cholangiocytes pCCA patients with available survival rates (*n* = 18) were divided into two groups; high GAL expressors and low GAL expressors. Remarkably, survival curves revealed significant differences in patient survival in this small cohort. Moreover, GAL expression showed a trend of inverse correlation with tumour grading, with high GAL expression in differentiated Grade 1 tumours and consecutively declining GAL expression with tumour dedifferentiation.

Regarding GAL receptors, GAL_1_-R was previously shown to mediate GAL-induced cholangiocellular hyperplasia during cholestasis [42,43]. Other studies assumed that GAL_1_-R signalling inhibits cell cycle progression [51]. In our study, GAL_1_-R was not upregulated in pCCA, but rather, it was higher in benign PIT, indicating that GAL_1_-R might cause anti-proliferative effects in CCA. Nevertheless, in the biliary vascular system of peritumoural tissues, we found significantly decreased GAL and GAL_3_-R expression in endothelial cells. In dermal inflammation, GAL is associated with vasoconstrictive effects, mediated by GAL_3_-R [44,58,59]. Therefore, reduced GAL and GAL_3_-R expression in biliary blood vessels might indirectly correlate with vasodilative effects and increased vascularisation in PIT. Furthermore, angiogenesis and advanced blood supply are two hallmarks of cancer [60]. Consequently, low levels of GAL in tumour tissues could lead to enhanced vascularisation, which can still be seen in neighbouring benign tissue. GAL_2_-R was also slightly reduced in tumour cells of pCCA. GAL_2_-R is associated with mitogenic and apoptotic signalling pathways [52,61]. It is suggested to initiate cell proliferation by activating ERK1/2 and AKT signalling [52]. ERK1/2 is also part of a cell signalling pathway in CCA that enhances proliferation and cell survival [2,3,8,42,62]. Moreover, a reduced GAL_2_-R mRNA level due to promoter methylation was observed in colorectal, prostate and breast cancer [35]. Therefore, the downregulation of GAL_2_-R in CCA could protect malignant cells from apoptotic signals and promote tumour development.

Low expression of GAL_3_-R was previously reported to correlate with enhanced metastasis and lymph node invasion in colorectal cancer [31]. We detected significantly lower GAL_3_-R expression levels in pCCA compared to healthy controls. Future studies are now warranted to validate this finding in larger patient cohorts. Detailed knowledge of the functions of the GAL system in CCA could open new avenues for the development of therapeutic strategies. In high GAL_3_-R expression in CCA tissues, GAL_3_-R-selective antagonists could be used as novel therapeutic agents. Another therapeutic option could be the use of radiolabelled cytotoxic agents linked to GAL_3_-R antagonists.

A limitation of this study is that the results were solely obtained via immunohistochemistry, as we intended to provide a first overview of the expression pattern of the GAL system in the healthy human biliary tract, cholestasis and CCA. A group with preneoplastic lesions was not included, but will be included in further studies comprised of larger patient cohorts.

## 5. Conclusions

To our knowledge, this is the first detailed characterisation of the expression pattern of the GAL system in the healthy human biliary duct. An additional analysis of cholangiocytes in cholestasis and pCCA revealed alterations in the expression of the GAL system that mainly affected the receptors. Remarkably, PIT and pCCA tissues primarily exhibited the downregulation of GAL and receptors compared to the controls. For PIT, the endothelial cells of blood vessels were primarily affected, and for pCCA, tissue alterations were found in cholangiocytes for GAL_3_-R (Figure 12). Our preliminary study on a small patient cohort revealed a potential direct correlation of GAL expression and an indirect correlation of GAL_3_-R expression with patient survival, which needs confirmation in a bigger sample cohort.

## Figures and Tables

**Figure 1 cells-12-01678-f001:**
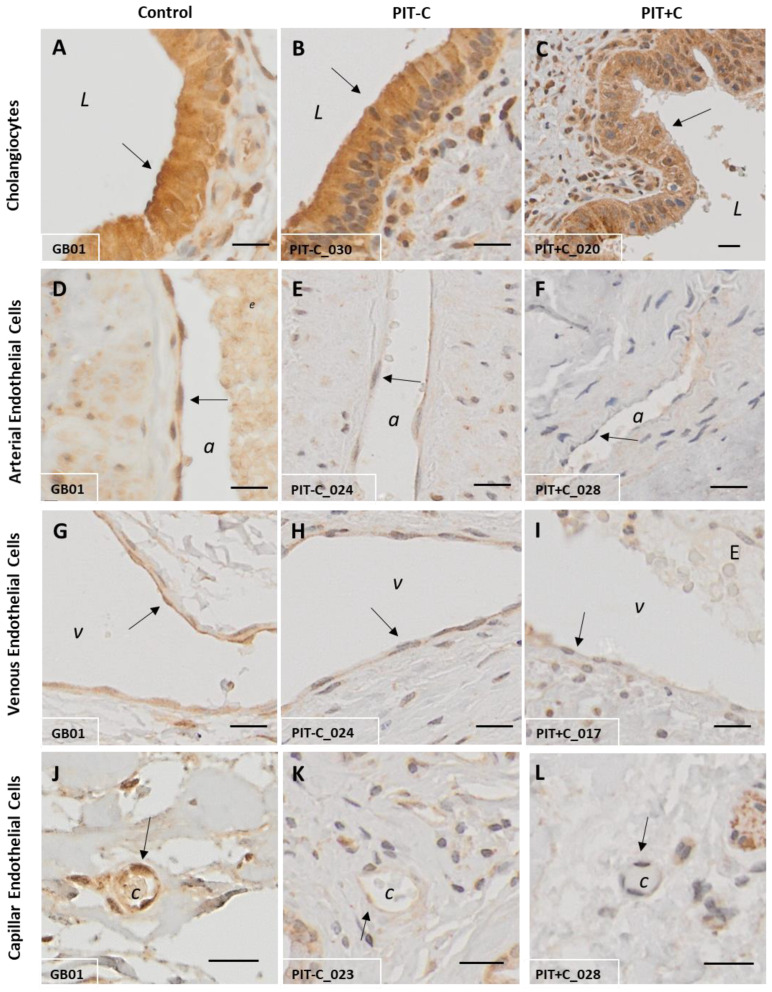
Representative IHC staining of GAL expression in control and peritumoural bile duct tissues. (**A**,**D**,**G**,**J**) Healthy control tissue; (**B**,**E**,**H**,**K**) PIT without cholestasis (PIT-C); (**C**,**F**,**I**,**L**) PIT with cholestasis (PIT+C). (**A**–**C**) Cholangiocytes; (**D**–**F**) arterial endothelium; (**G**–**I**) venous endothelium; (**J**–**L**) capillary endothelium. a = artery; v = vein; c = capillary; L = duct lumen; E = erythrocytes. Arrows point to structures of interest. Antibody: α-human GAL IgG. Scale bar = 20 µm. Characteristics of healthy individuals and patient samples are provided in Appendix A, and score values in Appendix A.

**Figure 2 cells-12-01678-f002:**
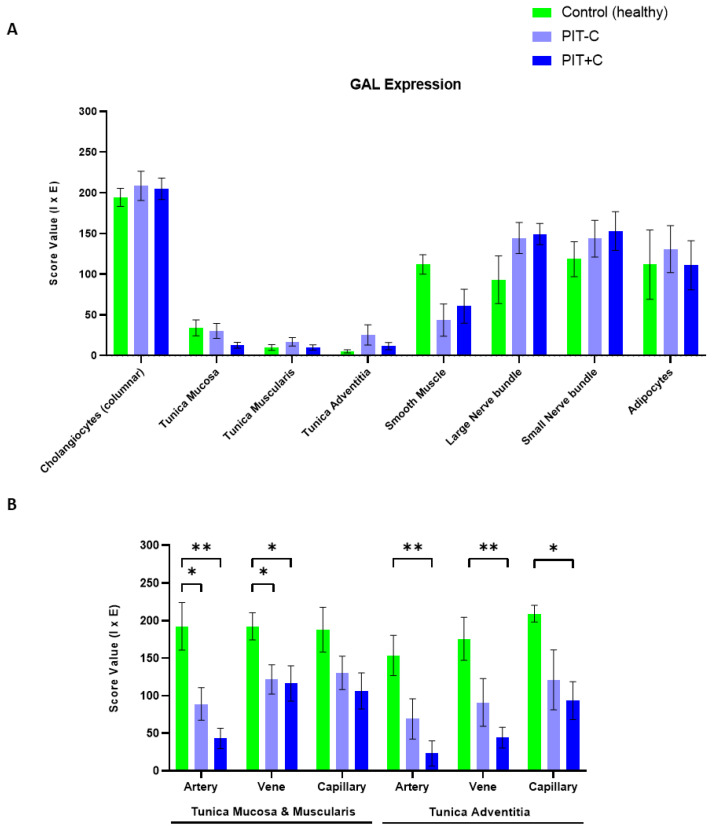
IHC score values of GAL expression in histology structures of control and peritumoural bile duct tissues with (PIT+C) and without cholestasis (PIT-C). (**A**) Score values of cholangiocytes, layers of connective tissue, smooth muscle, nerve fibres and fatty tissue. (**B**) Score values of endothelial cells in arterial, venous and capillary vessels, embedded in tunica mucosa and muscularis or tunica adventitia of bile duct cross sections. Control tissues derived from healthy individuals. PIT obtained from pCCA patients with or without cholestasis. Score value = mean ± SEM. Mann–Whitney-Test. * *p* < 0.05; ** *p* < 0.01.

**Figure 3 cells-12-01678-f003:**
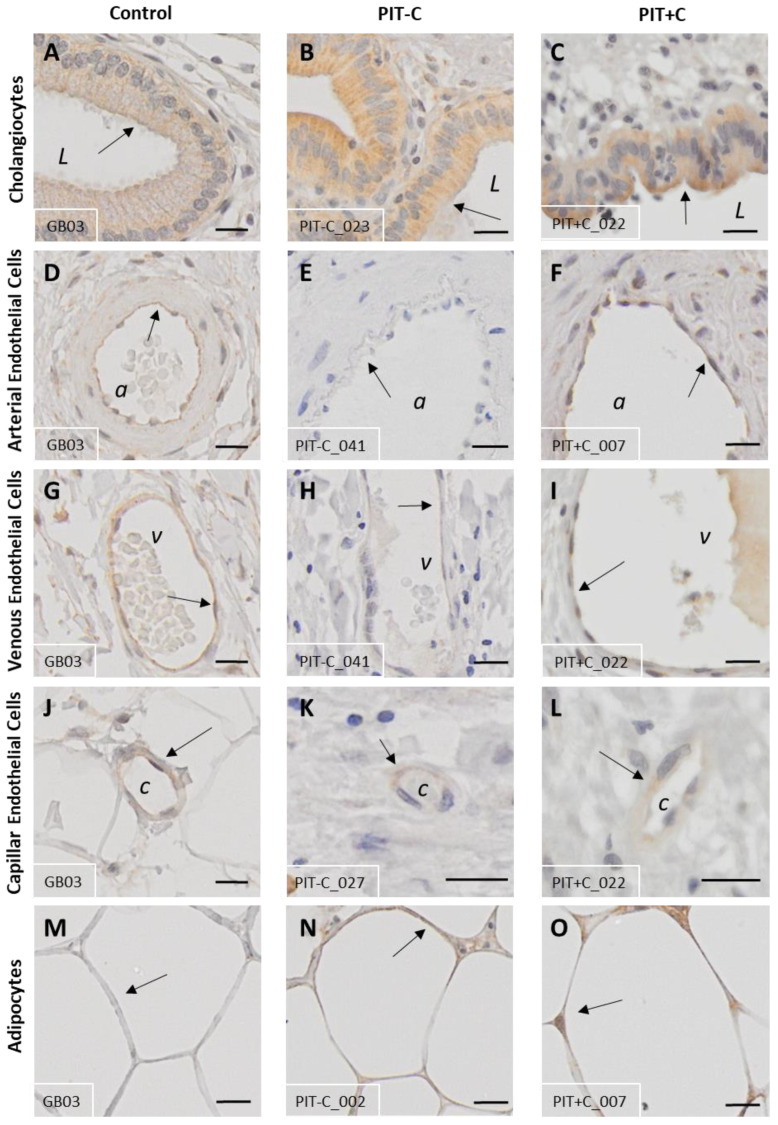
Representative IHC staining of GAL_1_-R expression in control and peritumoural bile duct tissues. (**A**,**D**,**G**,**J**,**M**) Healthy control tissue; (**B**,**E**,**H**,**K**,**N**) PIT without cholestasis (PIT-C); (**C**,**F**,**I**,**L**,**O**) peritumoural tissue with cholestasis (PIT+C). (**A**–**C**) Cholangiocytes; (**D**–**F**) arterial endothelium; (**G**–**I**) venous endothelium; (**J**–**L**) capillary endothelium; (**M**–**O**) adipocytes. a = artery; v = vein; c = capillary; L = duct lumen. Arrows point to structures of interest. Antibody: α-human GAL_1_-R IgG. Scale bar = 20 µm. Characteristics of healthy individuals and patient samples are provided in Appendix A, and score values in Appendix A.

**Figure 4 cells-12-01678-f004:**
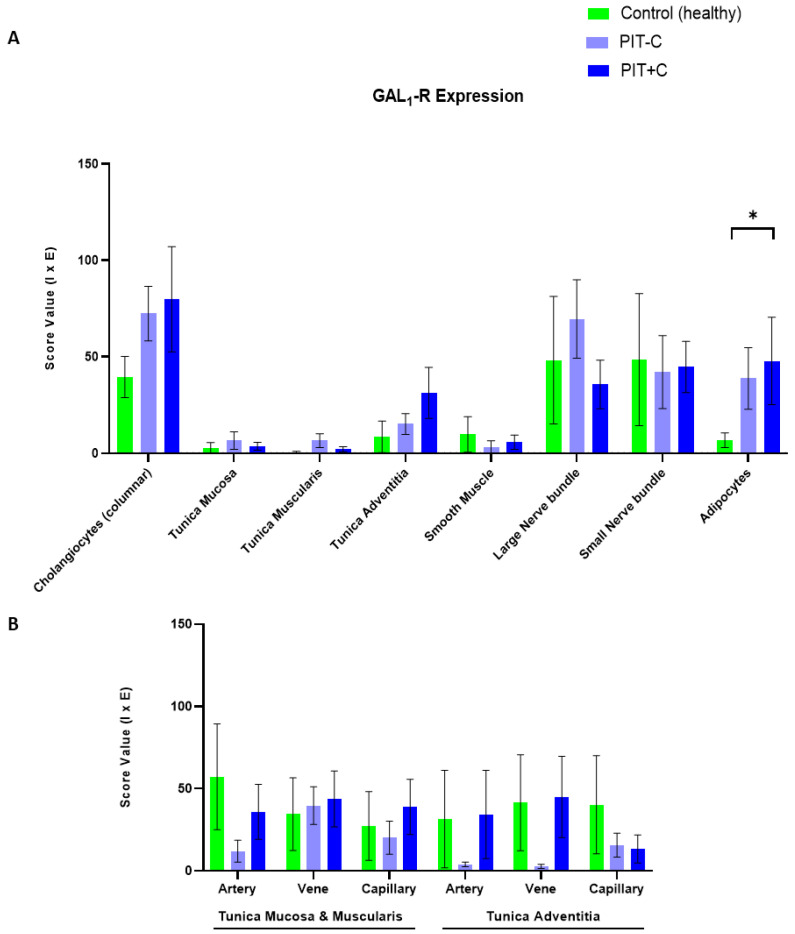
IHC score values of GAL_1_-R expression in histology structures of control and peritumoural bile duct tissues with (PIT+C) and without (PIT-C) cholestasis. (**A**) Scoring of cholangiocytes, layers of connective tissue, smooth muscle, nerve fibres and fatty tissue. (**B**) Scoring of endothelial cells in arterial, venous and capillary vessels, embedded in tunica mucosa and muscularis or tunica adventitia of bile duct cross sections. Control tissues derived from healthy individuals. PIT obtained from pCCA patients with or without cholestasis. Score value = mean ± SEM. * *p* < 0.05.

**Figure 5 cells-12-01678-f005:**
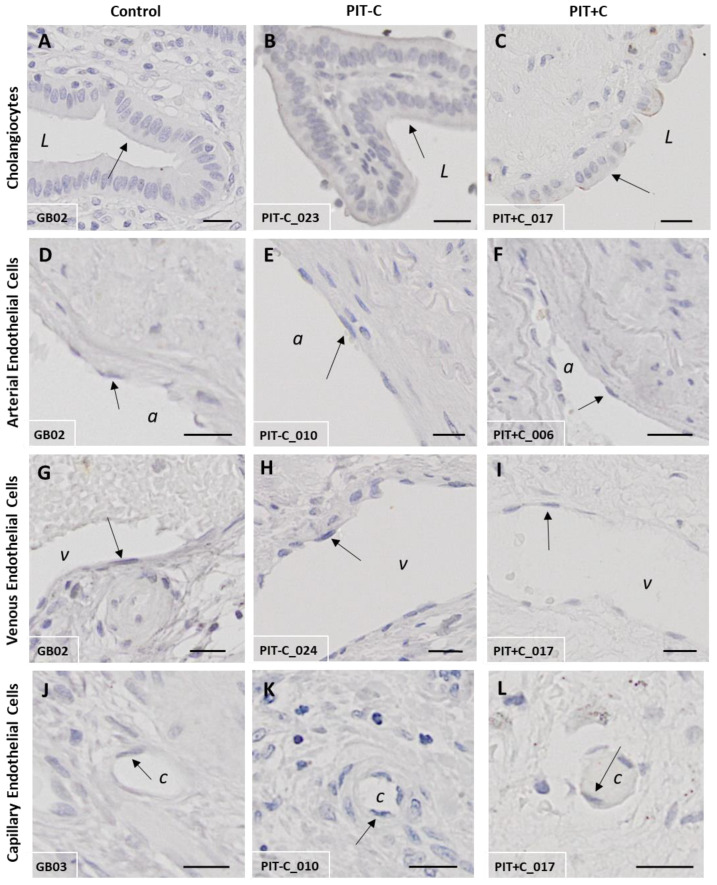
Representative IHC staining of GAL_2_-R in control and peritumoural bile duct tissues. (**A**,**D**,**G**,**J**) Healthy control tissue; (**B**,**E**,**H**,**K**) PIT without cholestasis (PIT-C); (**C**,**F**,**I**,**L**) PIT with cholestasis (PIT+C). (**A**–**C**) Cholangiocytes; (**D**–**F**) arterial endothelial cells; (**G**–**I**) venous endothelial cells; (**J**–**L**) capillary endothelial cells. a = artery; v = vein; c = capillary; L = duct lumen. Arrows point to structures of interest. Antibody: rabbit α-human GAL_2_-R IgG. Scale bar = 20 µm. Characteristics of healthy individuals and patient samples are provided in Appendix A, and score values in Appendix A.

**Figure 6 cells-12-01678-f006:**
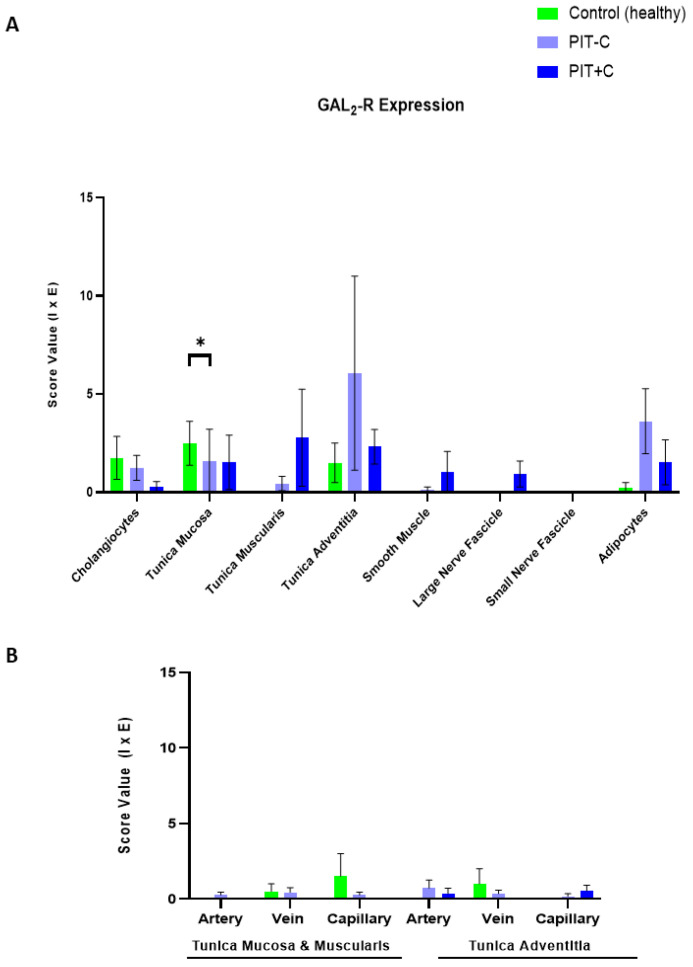
IHC score values of GAL_2_-R expression in tissues of control and peritumoural bile duct tissues with (PIT+C) and without (PIT-C) cholestasis. (**A**) Scoring of cholangiocytes, layers of connective tissue, smooth muscle, nerve fibres and fatty tissue. (**B**) Scoring of endothelial cells in arterial, venous and capillary vessels, embedded in tunica mucosa and muscularis or tunica adventitia of bile duct cross sections. Control tissues derived from healthy individuals. PIT obtained from pCCA patients with or without cholestasis. Score value = mean ± SEM. * *p* < 0.05.

**Figure 7 cells-12-01678-f007:**
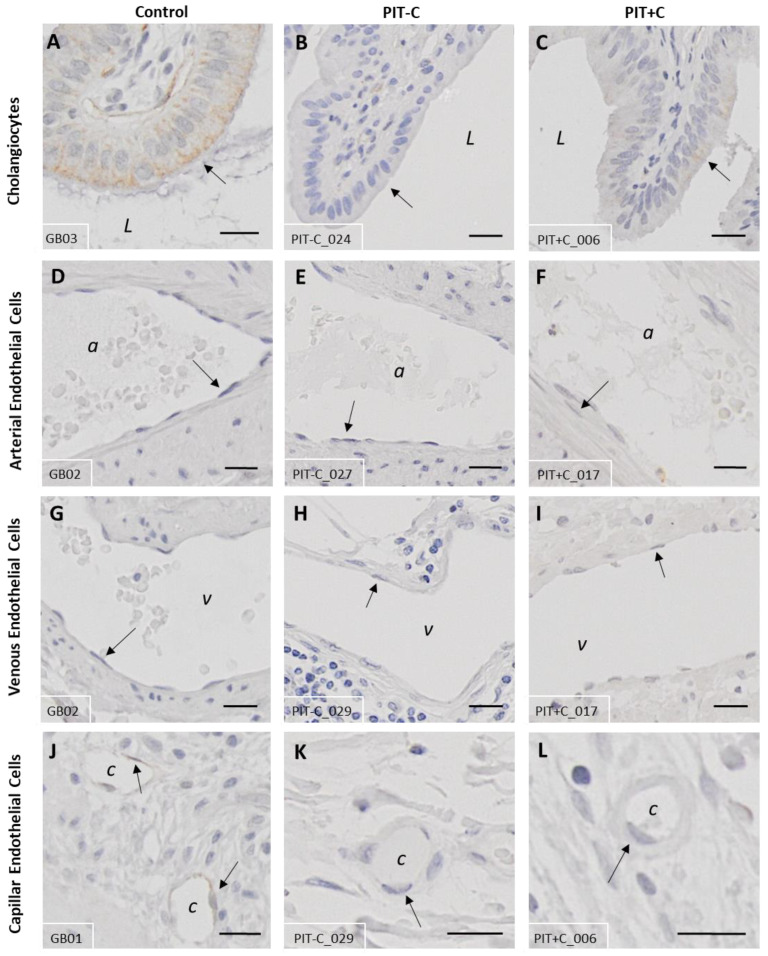
Representative IHC staining of GAL_3_-R expression in control and peritumoural bile duct tissue. (**A**,**D**,**G**,**J**) Healthy control tissue; (**B**,**E**,**H**,**K**) PIT without cholestasis (PIT-C); (**C**,**F**,**I**,**L**) PIT with cholestasis (PIT+C). (**A**–**C**) Cholangiocytes; (**D**–**F**) arterial endothelium; (**G**–**I**) venous endothelium; (**J**–**L**) capillary endothelium. a = artery; v = vein; c = capillary; L = duct lumen. Arrows point to structures of interest. Antibody: rabbit α-human GAL_3_-R IgG. Scale bar = 20 µm. Characteristics of healthy individuals and patient samples are provided in Appendix A, and score values in Appendix A.

**Figure 8 cells-12-01678-f008:**
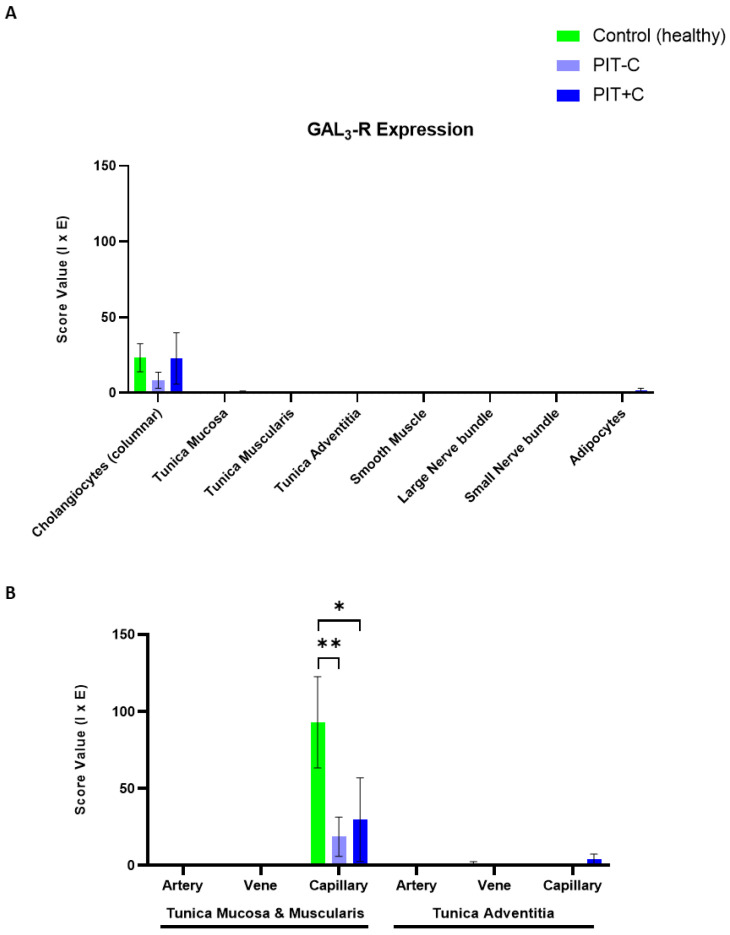
IHC score values of GAL_3_-R expression in histology structures of control and peritumoural bile duct tissues with (PIT+C) and without (PIT-C) cholestasis. (**A**) Scoring of cholangiocytes, layers of connective tissue, smooth muscle, nerve fibres and fatty tissue. (**B**) Scoring of endothelial cells in arterial, venous and capillary vessels, embedded in tunica mucosa and muscularis or tunica adventitia of bile duct cross sections. Control tissues derived from healthy individuals. PIT obtained from pCCA patients with or without cholestasis. Score value = mean ± SEM. * *p* < 0.05; ** *p* < 0.01.

**Figure 9 cells-12-01678-f009:**
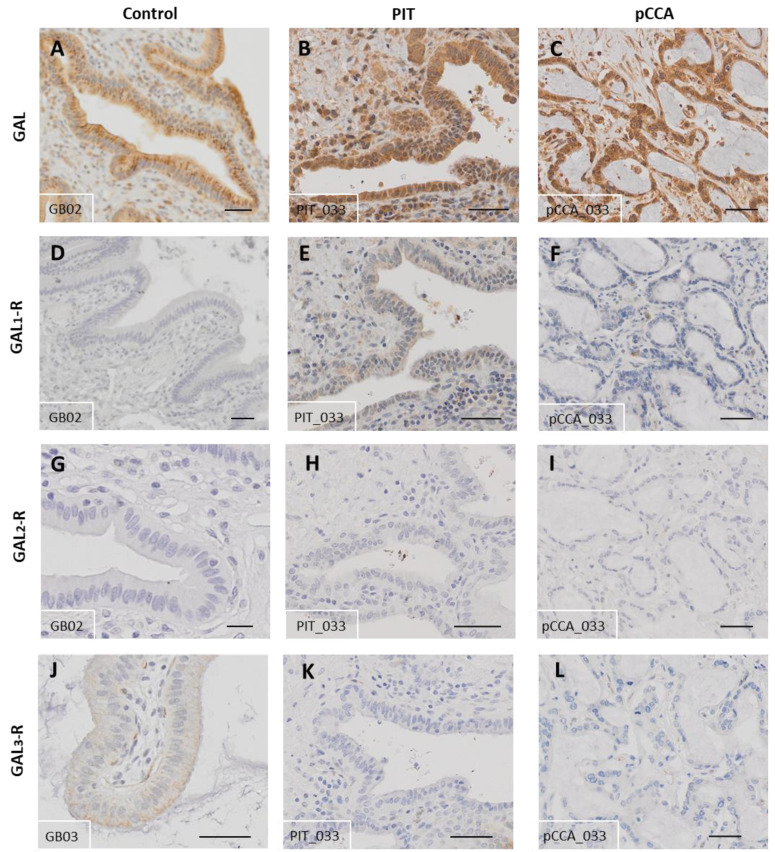
Representative immunohistochemical staining of pCCA (pCCA_033) in comparison to control (healthy) and PIT (PIT_033). (**A**–**C**) GAL; (**D**–**F**) GAL_1_-R; (**G**–**I**) GAL_2_-R; (**J**–**L**) GAL_3_-R. Antibody: (**A**–**C**) rabbit α-human GAL IgG; (**D**–**F**) rabbit α-human GAL_1_-R IgG; (**G**–**I**) rabbit α-human GAL_2_-R IgG; (**J**–**L**) rabbit α-human GAL_3_-R IgG. Control tissues derived from healthy individuals. PIT obtained from pCCA patients with or without cholestasis. Scale bar = 50 µm. Characteristics of healthy individuals and patient samples are provided in Appendix A, and score values in Appendix A.

**Figure 10 cells-12-01678-f010:**
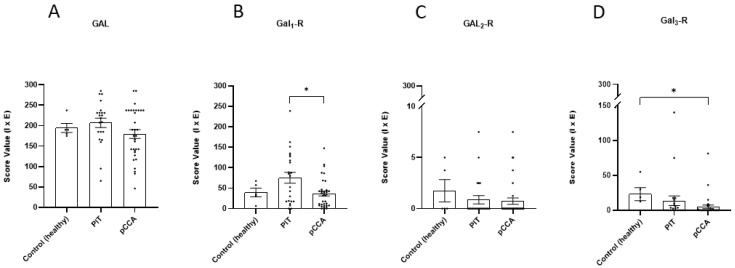
IHC score values of GAL and GAL_1–3_-R expression in cholangiocytes/tumour cells, lining healthy and pathological bile ducts (PIT, pCCA). (**A**) GAL expression; (**B**) GAL_1_-R expression; (**C**) GAL_2_-R expression; (**D**) GAL_3_-R expression. Control tissues derived from healthy individuals. PIT and pCCA obtained from pCCA patients. Score value = mean ± SEM. * *p* < 0.05.

**Figure 11 cells-12-01678-f011:**
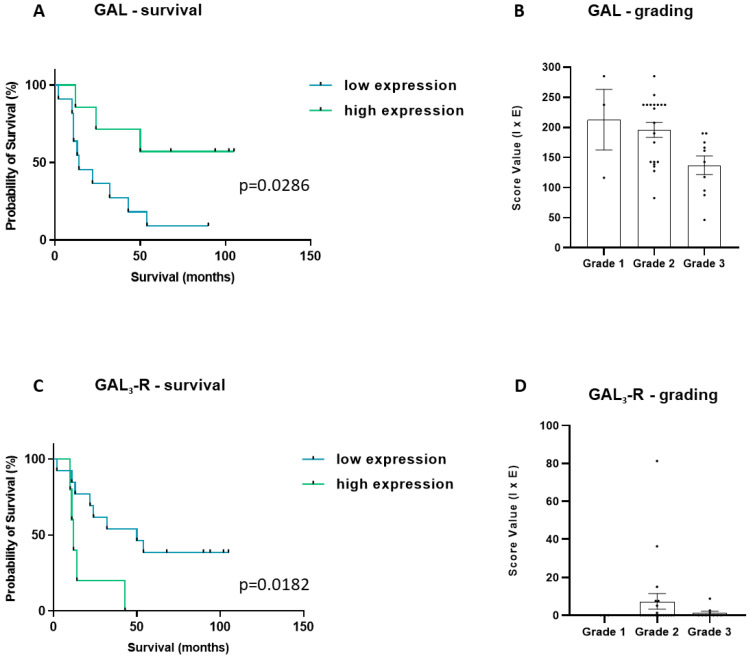
Kaplan–Meier curves of pCCA patients exhibiting low or high (**A**) GAL or (**B**) GAL_3_-R score values in cholangiocytes. High expressors are shown in green and low expressors in blue. (**A**) GAL: high expressors: *n* = 7, low expressors: *n* = 11. (**B**) GAL expression in cholangiocytes of pCCA patients and associated grading. Comparison of mean GAL expression tended to inversely correlate with tumour grading. (**C**) GAL_3_-R: high expressors: *n* = 5, low expressors: *n* = 13. (**D**) GAL_3_-R expression in cholangiocytes of pCCA patients and associated grading. (**B**,**D**) Grade 1: *n* = 3, Grade 2: *n* = 21, Grade 3: *n* = 10. Both depicted as scatter dot plots.

**Figure 12 cells-12-01678-f012:**
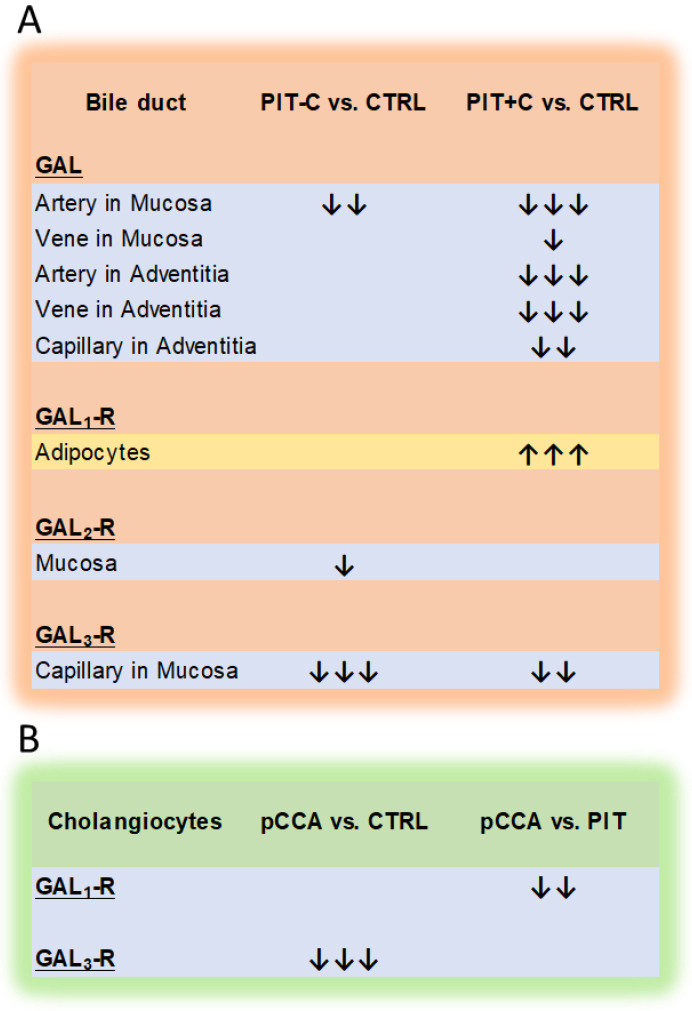
Schematic illustration of significant alterations in the expression of the GAL system in the human bile duct. (**A**) Comparison of mean expression of PIT-C and PIT+C versus healthy controls (CTRLs). (**B**) Comparison of mean expression in cholangiocytes of pCCA with healthy tissue (CTRLs) and PIT. ↑ upregulation and ↓ downregulation compared to control; one arrow represents a relative difference of ≥25%, two arrows ≥ 50%, and three arrows ≥ 75%.

**Table 1 cells-12-01678-t001:** Overview of samples derived from healthy individuals and pCCA patients.

Origin	Sample Group	Abbreviation	Sample Size (*n*)
Healthy patient	Control tissue	Control	5
pCCA patient	Peritumoural without cholestasis	PIT-C	10
	Peritumoural with cholestasis	PIT+C	10
	Klatskin tumour	pCCA	33

In Control, PIT-C and PIT+C tissue sections, the following cell types were analysed for GAL/GAL_1–3_-R expression: cholangiocytes, mucous glands, fibroblasts, nerve fibres, arterial and venous endothelium, capillary endothelium, adipocytes and smooth muscle. In the samples of pCCA, solely tumour cells were eligible for analysis.

**Table 2 cells-12-01678-t002:** Validated antibodies for the human galanin system [44].

Antibody	Species	Dilution *	Company	Order No.
GAL	rabbit	1:300	Peninsula	T-4325
GAL_1_-R	rabbit	1:200	Genetex	GTX108207
GAL_2_-R	rabbit	1:400	PTG-Lab	S4510-1
GAL_3_-R	rabbit	1:500	Genetex	GTX108163

* Dilution in Dako antibody diluent with background-reducing components (Agilent Technologies, Glostrup, Denmark).

**Table 3 cells-12-01678-t003:** Overview of the expression of the human galanin system in healthy controls and peritumoural tissue with and without cholestasis.

	Score Values			Comparison of PIT-C VS. Ctrl		Comparison of PIT+C VS. Ctrl				
Tissue	Control Mean ± SEM	PIT-C Mean ± SEM	PIT+C Mean ± SEM	Rel. Diff.	*p* Value	Rel. Diff.	*p* Value	Subcellular Localisation		
**GAL**										
Ch	195 ± 11	209 ± 18	205 ± 13	7%	0.2956	5%	0.9770	Plasma membrane, cytoplasm		
Muc	34 ± 10	30 ± 9	13 ± 4	12%	0.5238	62%	0.0622			
Mus	10 ± 4	17 ± 5	10 ± 3	70%	0.7928	0%	nc			
Adv	5 ± 2	25 ± 12	12 ± 5	400%	0.4066	140%	0.3144			
Musc	112 ± 12	44 ± 20	61 ± 21	61%	0.0622	46%	0.1255			
LNB	93 ± 29	145 ± 20	149 ± 13	56%	0.2795	60%	0.2222			
SNB	119 ± 22	144 ± 23	153 ± 24	21%	0.4409	29%	0.4584			
Adi	112 ± 43	131 ± 29	111 ± 30	17%	0.8227	0.9%	>0.9999			
ArtMuc	192 ± 32	89 ± 22	43 ± 14	54%	**0.0420**	78%	**0.0025**			
VenMuc	192 ± 18	122 ± 20	116 ± 24	36%	0.0383	40%	**0.0303**			
CapMuc	188 ± 30	130 ± 22	106 ± 24	31%	0.2166	44%	0.0759			
ArtAdv	154 ± 27	69 ± 27	23 ± 17	55%	0.0694	85%	**0.0051**			
VenAdv	176 ± 29	91 ± 32	44 ± 14	48%	0.1061	75%	**0.0051**			
CapAdv	209 ± 11	121 ± 40	93 ± 25	42%	0.1190	56%	**0.0152**			
**GAL_1_-R**										
Ch	40 ± 11	72 ± 14	80 ± 27	80%	0.3808	100%	0.5185	Plasma membrane, cytoplasm		
Muc	3 ± 3	7 ± 5	4 ± 2	133%	>0.9999	33%	0.8352	Plasma membrane		
Mus	1 ± 1	7 ± 4	2 ± 1	600%	0.6478	100%	0.5431	Plasma membrane		
Adv	9 ± 8	15 ± 5	31 ± 13	67%	0.4180	244%	0.0808			
Musc	10 ± 9	3 ± 3	6 ± 4	70%	0.4322	40%	>0.9999			
LNB	48 ± 33	70 ± 20	36 ± 13	46%	0.5462	25%	0.9138			
SNB	49 ± 34	42 ± 19	45 ± 13	14%	0.7558	8%	0.5689			
Adi	7 ± 4	39 ± 16	48 ± 23	457%	0.1916	586%	**0.0341**			
ArtMuc	57 ± 32	12 ± 7	36 ± 17	79%	0.2692	37%	0.7790			
VenMuc	35 ± 22	40 ± 11	44 ±17	14%	0.8759	26%	0.9264			
CapMuc	27 ± 21	20 ± 10	39 ± 17	26%	0.1696	44%	0.7208			
ArtAdv	32 ± 30	4 ± 1	34 ± 27	88%	0.7727	6%	0.5707			
VenAdv	42 ± 29	3 ± 1	45 ± 25	93%	0.2992	7%	>0.9999			
CapAdv	40 ± 30	16 ± 7	13 ± 9	60%	0.8377	68%	0.6212			
**GAL_2_-R**										
Ch	2 ± 1	1 ± 1	0	50%	0.7663	100%	0.3407	Plasma membrane		
Muc	3 ± 1	2 ± 2	2 ± 1	33%	**0.0374**	33%	0.1938			
Mus	0	0	3 ± 2	0%	nc	nd	0.2582			
Adv	2 ± 1	6 ± 5	2 ± 1	200%	>0.9999	0%	nc			
Musc	0	0	1 ± 1	0%	nc	nd	>0.9999			
LNB	0	0	1 ± 1	0%	nc	nd	0.4872			
SNB	0	0	0	0%	nc	0%	nc			
Adi	0	4 ± 2	2 ± 1	nd	0.2328	nd	0.6703			
ArtMuc	0	0	0	0%	nc	0%	nc			
VenMuc	1 ± 1	0	0	100%	>0.9999	100%	0.4167			
CapMuc	2 ± 2	0	0	100%	0.6870	100%	0.3571			
ArtAdv	0	1 ± 1	0	nd	0.4697	0%	nc			
VenAdv	1 ± 1	0	0	100%	>0.9999	100%	>0.9999	Plasma membrane		
CapAdv	0	0	1 ± 0	0%	nc	nd	0.4697			
**GAL_3_-R**										
Ch	23 ± 9	8 ± 5	23 ± 17	65%	0.0799	0%	nc	Plasma membrane		
Muc	0	0	1 ± 1	0%	nc	nd	>0.9999			
Mus	0	0	0	0%	nc	0%	nc			
Adv	0	0	0	0%	nc	0%	nc			
Musc	0	0	0	0%	nc	0%	nc			
LNB	0	0	0	0%	nc	0%	nc			
SNB	0	0	0	0%	nc	0%	nc			
Adi	0	0	2 ± 2	0%	nc	nd	>0.9999			
ArtMuc	0	0	0	0%	nc	0%	nc			
VenMuc	0	0	0	0%	nc	0%	nc			
CapMuc	93 ± 30	19 ± 13	30 ± 27	80%	**0.0030**	68%	**0.0326**			
ArtAdv	0	0	0	0%	nc	0%	nc			
VenAdv	2 ± 1	0	0	100%	0.1282	100%	0.1515			
CapAdv	0	0	4 ± 4	0%	nc	nd	0.4697			

Abbreviations: Chc, cholangiocytes; Muc, mucosa; Mus, muscularis; Adv, adventitia; Musc, smooth muscle; LNB, large nerve bundle; SNB, small nerve bundle; Adi, adipocytes; ArtMuc, artery in mucosa; VenMuc, vein in mucosa; CapMuc, capillary in mucosa; ArtAdv, artery in adventitia; VenAdv, vein in adventitia; CapAdv, capillary in adventitia; PIT, peritumoural tissue; Rel. Diff., relative difference; SEM, Standard Error of the Mean; Ctrl, Control; significant differences between study groups are marked in bold; nd, not determinable; nc, not computable.

**Table 4 cells-12-01678-t004:** Overview of the expression of the human GAL system in healthy controls, peritumoural tissue and pCCA tissue.

	Score Values			Comparison of pCCA VS. Ctrl		Comparison of pCCA VS. PIT		
Staining	Control Mean ± SEM	PIT Mean ± SEM	pCCA Mean ± SEM	Rel. Diff.	*p* Value	Rel. Diff.	*p* Value	Subcellular Localisation
**GAL**	195 ± 11	207 ± 55	180 ± 11	8%	>0.9999	13%	0.3301	Plasma membrane, cytoplasm
**GAL_1_-R**	40 ± 11	75 ± 13	37 ± 6	8%	>0.9999	51%	**0.0459**	Plasma membrane, cytoplasm
**GAL_2_-R**	2 ± 1	1 ± 0	1 ± 0	50%	0.4680	0%	>0.9999	Plasma membrane
**GAL_3_-R**	23 ± 9	13 ± 7	5 ± 3	78%	**0.0120**	62%	0.1280	Plasma membrane

Abbreviations: PIT, peritumoural tissue; Rel., relative; Diff., difference; SEM, Standard Error of the Mean; pCCA, perihilar cholangiocarcinoma tissue; Ctrl, Control; significant differences between study groups are marked in bold.

## Data Availability

Not applicable.

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
