# Peer review of "Galanin System in the Human Bile Duct and Perihilar Cholangiocarcinoma"

_cells, 2023, doi:10.3390/cells12131678_

Round 1

Reviewer 1 Report

The paper by Fitzner et al. aims to characterize the GAL system in perihilar cholangiocarcinoma versus nontumor tissue of the healthy bile duct or surrounding nontumor tissue. The poor prognosis of biliary neoplasm and the lack of therapeutic strategies, with the exception of surgery in resectable disease, prompt the investigation of candidate early markers. The Fitzner manuscript identifies an interesting issue and is a descriptive work that represents a preliminary attempt to histomorphologically characterize the expression of the GAL system in the context of biliary neoplasms and surrounding tissue.

However, the extrapolation of purely descriptive data into terms of clinical significance is difficult to achieve and the attempt by Fitzner et al. does not appear entirely convincing.

Here are the main issues that deserve careful consideration by the authors.

1) The authors should better define what they mean by “surrounding non-tumor tissue”. Pre-neoplastic biliary lesions with crowded epithelial cells or papillary mucosa shape are often present along a diseased bile duct and show altered biology. Some photomicrographs of BO-KlGa and BC-KlGa may not be easily assigned to a completely normal biliary mucosa. Furthermore, it could be very interesting to extend the analysis of GAL system expression to include some biliary preneoplastic lesions.

2) Considering that the approach of the work is merely descriptive only at the immunohistochemical level, it would be useful to confirm/strengthen the results by associating the analysis of some further intracellular factor or mediator directly connected to the activation, for example of the different forms of GAL-R. In this way, the effective activation of the GAL system and, consequently, its presumed correlation with the pathogenetic mechanism of the diseased tissue could be better investigated.

3) On the other hand, investigating the expression of some regulatory system upstream of the expression of the GAL system could be equally useful for corroborating the data. As the same authors mention, epigenetic factors regulating the expression of the GAL system could be implicated.

4) Survival data are certainly stimulating but should be extracted from a much larger case series, also considering that various data are missing in the cohort presented by the authors, greatly reducing the size of the sample that can actually be used for this analysis. If the authors still want to present these data, they need to downplay their importance, best discussing them as merely preliminary and essentially stripping them from the beginning of the paper.

5) The immunohistochemical data are difficult for the reader to follow and some summary tables could better provide a complete final view of the main results. A simplified naming of the different study groups should also be prepared.

The overall quality of English is acceptable and only a few spelling errors should be amended.

Author Response

Please see in the attachement

Reviewer 2 Report

Authors have performed an interesting and well organized study focused on the expression of the galaninergic system  in the human bile duct and perihilar cholangiocarcinoma. They have applied an immunohistochemical  technique to study the expressions of galanin and galanin 1, 2 and 3 receptors in control patients (n = 5), in those suffering from benign peritumoral bile duct (with or without cholestasis)(n = 20) and in patients with perihilar cholangiocarcinoma (n = 33).  Moreover, they state that galanin and galanin 3 receptor expressions in cholangiocytes represent potential biomarkers for survival in patients suffering from perihilar cholangiocarcinoma.

I suggest the following points to be improved:

1.      Lines 20 and 54: peptide instead of neuropeptide. Check it along the text.

2.      Photographs of the negative controls performed must be shown.

3.      Explain clearly in the text where the immunoreactivity was observed (nuclei, cytoplasm, plasma membrane) in each cell type. This information is lacking in some cell types. According to some photographs, it seems that the immunoreactivity is located in the nuclei (e.g, endothelial cells, adipocytes). If this is the case, discuss this finding.

4.      Figure 1J is out of focus and the tissue is not well conserved. Replace it and show, if it is possible, an improved photograph.

5.      A comparative Table including all the results obtained (e.g., groups studied, cell types) will help to have a better idea regarding the important findings reported. Include in this Table what is indicated in point 3.

6.      Discussion: Could the authors suggest some therapeutic strategies (e.g., targeted radionuclide cancer therapy) to treat the disease according to the findings reported?

7.      Conclusion. A Figure summarizing the findings reported and including some ideas/hypothesis mentioned in Discussion (e.g., signalling pathways, proliferative/antiproliferative action, epigenetic mechanisms) would be welcome. Indicate in this section or in Discussion the main research lines that must be developed in the future regarding the topic reviewed.

8.      References. For example reference 9, title: Molecular pathogenesis of cholangiocarcinoma instead of Molecular Pathogenesis of Cholangiocarcinoma. Check the list of references.

     In sum, it is a well performed study which remarkably increases the knowledge on the galaninergic system in the human bile duct.

Reviewer 3 Report

1.     At least Figure 1 and Figure 2 show a high background staining, which would be suspicious for an unspecific staining effect and therefore would not qualify for a valid expression of this marker. Please discuss this (methods, discussion), did you validate the antibodies (gene knockout, at least comparison tests with plausible negative tissue, if so, describe previous tests), is there any literature on validation of antibodies applied? 

2.     Please provide a tabular overview of your comparison tests including p-values.

3.     Please add a comment on limitations of your study, e.g. small sample size and implications.

Round 2

Reviewer 1 Report

The authors significantly improved the quality of the manuscript.

Reviewer 3 Report

The manuscript has been significantly improved.